# 1,5,6-Trimethoxy-2,7-dihydroxyphenanthrene from *Dendrobium officinale* Exhibited Antitumor Activities for HeLa Cells

**DOI:** 10.3390/ijms242015375

**Published:** 2023-10-19

**Authors:** Chong Liang, Chonglun Zhang, Yinlin Zhuo, Baocheng Gong, Weizhuo Xu, Guogang Zhang

**Affiliations:** 1School of Traditional Chinese Materia Medica, Shenyang Pharmaceutical University, 103 Wenhua Road, Shenhe District, Shenyang 110016, China; liangchong7628@163.com (C.L.); 18750097073@163.com (Y.Z.); gkcmu_1h@163.com (B.G.); 2School of Life Sciences and Biopharmaceuticals, Shenyang Pharmaceutical University, 103 Wenhua Road, Shenhe District, Shenyang 110016, China; zhangcl0630@163.com; 3School of Functional Food and Wine, Shenyang Pharmaceutical University, 103 Wenhua Road, Shenhe District, Shenyang 110016, China

**Keywords:** *Dendrobium officinale*, phenanthrene, 1,5,6-trimethoxy-2,7-dihydroxyphenanthrene, P53, cytotoxic

## Abstract

Natural products are irreplaceable reservoirs for cancer treatments. In this study, 12 phenanthrene compounds were extracted and isolated from *Dendrobium officinale*. Each chemical structure was identified using comprehensive NMR analysis. All compounds were evaluated for their cytotoxic activities against five tumor cell lines, i.e., HeLa, MCF-7, SK-N-AS, Capan-2 and Hep G2. Compound 5, 1,5,6-trimethoxy-2,7-dihydroxyphenanthrene, displayed the most significant cytotoxic effect against HeLa and Hep G2 cells, with an IC_50_ of 0.42 and 0.20 μM. For Hela cells, further experiments demonstrated that compound 5 could obviously inhibit cell migration, block cell cycle in the G0/G1 phase and induce apoptosis. Expression measurements for p53 indicated that knock down of p53 by siRNA could mitigate the apoptosis induced by compound 5. Therefore, the compound 5 is a potential candidate drug for HeLa cells in cervical cancer.

## 1. Introduction

Cancer is one of the major health problems in the world [1]. According to the report of the International Agency for Research on Cancer [2], in 2020, there were 19.3 million new cancer cases and nearly 10 million cancer deaths worldwide. Cervical cancer is the second most common cancer in women worldwide [3]. Despite surgery, chemotherapy and radiation therapy, cervical cancer still presents a high rate of aggressiveness and mortality [4]. 

Multiple natural products have exhibited activities against cervical cancer, such as benzylcinnamide [5], resveratrol [6], berberine [7], *Ficus religiosa* extracts [8], *Azadirachta indica* extracts [9], green tea compounds [10], polyphenols and flavonoids [11].

Orchidaceae family-derived natural products have also been applied for their antitumor properties for years around the world. As a globally grown flowering plant, orchidaceae has approximately 35,000 species [12]. Previous literature has reported that compounds from *Dendrobium* demonstrated cytotoxic activity against cervical cells [13,14,15,16,17]. Moreover, *Dendrobium chrysanthum* [18] and *Dendrobium venustum* [19] also exhibited antitumor activities against HeLa cells.

*Dendrobium officinale* Kimura et Migo, a Chinese endemic herb plant, is a member of the Dendrobium plants (Orchidaceae), which are mainly distributed in eastern and southeastern China [20]. Among phenanthrene compounds from the *Dendrobium* plants, bletilols [21] and pholidonone [22] were cytotoxic to cancer cells. Dendrobine and its derivatives have been deeply studied for their apoptotic capacities and signaling pathway [23,24]. Phenanthrenequinones [25] and phenanthrene [26] may induce cell cycle arrest and apoptosis and inhibit cell migration. 9,10-dihydrophenanthrene and bibenzyl enantiomers from *Bletilla striata* [27], biphenanthrenes from *Cremastra appendiculata* [28], other dihydrophenanthrenes from *Pholidota cantonensis* [29], 1,4-dihydrophenanthrenequinone phocantone [30] and phenanthrenes from *Cymbidium faberi* [31] also displayed cytotoxic activities. However, few reports could be found for the evaluation of the antitumor characteristics of extracts of *Dendrobium officinale*.

In this study, twelve phenanthrene compounds were isolated and identified from the stem of Dendrobium officinale. Extensive research was performed on these isolates’ activities against HeLa, MCF-7, SK-N-AS, Capan-2 and HepG-2 cells.

## 2. Results

### 2.1. Extraction and Identification of Compounds 1–12 from Dendrobium officinale

The purification of the 95% EtOH extract of the stem from *Dendrobium officinale* using chromatographic methods yielded twelve known phenanthrenes. The structures of compounds 1–12 were characterized via extensive spectroscopic analyses including ^1^HNMR and ^13^CNMR (Figure 1 and Appendix A).

The twelve known phenanthrenes (1–12) were identified as 4,4’,7,7’-tetrahydroxy-2,2’,8,8’-tetramethoxy-1,1’-diphenanthrene (1) [32], densiflorol B (2) [33], 4-methoxy-2,5-dihydroxyphenathrene (3) [34], 2,7-dihydroxy-3,4-dimethoxyphenanthrene (4) [35], 1,5,6-trimethoxy-2,7-dihydroxyphenanthrene (5) [36], 3,7-dihydroxy-2,4-dimethoxyphenanthrene (6) [36], orchinol (7) [37], erianthridin (8) [36], ephemeranthol A (9) [38], Phoyunnanin C (10) [37], 2,4,7-trihydroxy-9,10-dihydrophenanthrene (11) [38] and 4-methoxy-9,10-dihydrophenanthrene-2,3,7-triol (12) [39] based on the analysis of their spectroscopic data and comparison with the reported literature.

### 2.2. MTT Assay

All 12 compounds were tested for cytotoxic activity against HeLa, MCF-7, SK-N-AS, Capan-2 and Hep G2 cells using the MTT method. As shown in Table 1, compound 1 displayed weak cytotoxic activity against all five cell lines, compound 2 showed strong cytotoxic activities against Capan-2 and Hep G2 cells with IC_50_ values of 14.96 and 10.87μM and compound 5 exhibited remarkable cytotoxic activity against HeLa and Hep G2 cells with IC_50_ values of 0.42 and 0.20 μM, which is almost 20 and 10 times lower than the IC_50_ values of cicplatin, 7.84 and 2.28 μM, and almost 10 times and approximately equal to the IC_50_ values of paclitaxel, 2.01 and 0.21 μM, respectively.

For compound 5 against HeLa cells, microscopic morphological data indicated that higher compound concentration caused significant cell shrinkage, therefore decreasing the cell survival rate. This is consistent with the apoptosis results. The survival rate curve of HeLa cells was decreased after the intervention of compound 5 for 12 h, 24 h and 48 h, respectively. These results indicated that compound 5 could significantly inhibit the proliferation of HeLa cells in a concentration-dependent and time-dependent manner. Therefore, compound 5 was used to further investigate the antitumor mechanism in HeLa cells (Figure 2).

### 2.3. Cell Confluence Rate Analysis

The cell confluence rate curve was performed with the IncuCyte living cell dynamic imaging system; after the analysis, the results showed that the cell fusion rate decreased gradually with the increase of compound 5 concentration (0–100 μM) (Figure 3). This result suggested that the compound 5 exhibited concentration-dependent proliferation inhibition on HeLa cells, which was consistent with the above MTT assay results.

### 2.4. Cell Migration Analysis

The cell scratch experiments were performed to evaluate the capacity of compound 5 towards HeLa cells. The experiment results showed that after 24 h of scratch, the migration distance of cells treated by compound 5 was shorter and the scratch gap was significantly wider than the DMSO control group. Compared with the control group, the cell migration rate of 0.84 μM (2 × C_50_) in the compound 5 treatment group was significantly higher (Figure 4).

### 2.5. Apoptosis Analysis

To perform the apoptosis analysis, Hoechst 33258 was used to provide blue fluorescence, which may indicate the progression of apoptosis. Figure 5A indicated the obvious HeLa apoptosis due to the treatment with compound 5 with increasing concentration. Then, single PI staining combined with flow cytometry was carried out to detect apoptosis (Figure 5B). The results showed that after the treatment with compound 5, the proportion of SubG1 phase cells increased, indicating that compound 5 could induce apoptotic cell death in HeLa cells. Then, AnnexinV-FITC/PI staining was used to measure the percentage of apoptotic cells. As shown in Figure 5C, the apoptosis rate increased from 7.15% in the control group to 93.46% (compound 5, 12.5 μM); thus, a concentration-dependent manner for the activity of compound 5 towards HeLa apoptosis was again identified. Furthermore, the apoptotic protein levels were evaluated via Western blot analysis; the elevated ratio of Bax/Bcl-2 (Figure 5D) also confirmed that compound 5 could induce apoptotic cell death in HeLa cells.

### 2.6. Cell Cycle Analysis

Flow cytometry with PI single stain was applied to detect the cell cycle of HeLa cells treated with compound 5. Those cells were incubated with compound 5 (3.125 μM) for 36 h. After 36 h of treatment, the proportion of the G0/G1phase increased, the proportion of the S phase decreased and the proportion of G2/M also decreased, indicating that compound 5 could block the cell cycle in the G0/G1phase (Figure 6).

### 2.7. Compound 5 Exerts Anti-HeLa Effect by Targeting P53

Much literatures has shown that the p53 gene exists in wild-type HeLa cells [6,7,8,9,10,11,12,13,14,15,16,17]. Based upon to the analysis of TCGA database [18], it could be summarized that high expression of p53 protein could significantly improve the survival rate of patients.

When HeLa cells were treated with different concentrations of compound 5, the p53 and MDM2 protein expression demonstrated a dose-dependent pattern (Figure 7).

To further investigate the role of p53 in compound 5-treated HeLa cells, the p53 gene was specifically knocked out using siRNA technology. Three candidate RNAi sequences were screened for their efficacy, and #3 RNAi sequence was applied for the final experiment. The experimental results showed that down regulation of p53 expression increased the confluence rate and survival rate of HeLa cells treated with compound 5 (Figure 8 and Figure 9). In other words, the anti-proliferative activity of compound 5 might be mitigated by p53.

## 3. Discussion

In this study, twelve phenanthrene compounds were obtained from the stems of *Dendrobium officinale* and their cytotoxic effect towards cervical cancer cells were screened. Among those compounds, compound 1 demonstrated weak inhibition activities for those five cell lines, while compound 5 exhibited the most significant cytotoxic effect against HeLa and Hep G2 cells, with IC_50_ values of 0.42 and 0.20 μM. This low concentration is quite exciting, exhibiting the same efficacy with as little as almost 1/20 and 1/10 of the concentration compared to cisplatin. Combined with other confluence and cell cycle inhibition data, it is reasonable to believe that compound 5 is an excellent antitumor candidate for the treatment of cervical cancer.

Ching-Ying Kuo et al. reported natural phenanthrenes’ antiproliferative potential in cervical cancer cell lines [40], Dehydrojuncuenin B and Jinflexin A exhibited promising cytotoxic activities towards Hela cells, with IC_50_ values of 16.57 and 17.38, while Juncuenin B, Dehydrojuncusol and Juncusol demonstrated better activities, with IC_50_ values of 2.9, 2.22 and 0.95. Changkang Li et al. reported that bibenzyl derivatives from *Dendrobium gratiosissimum* exhibited IC_50_ values of 6.25~22.06 towards gastric carcinoma cell line BGC823 [41]. In Dóra Stefkó et al.’s research, several phenanthrenes from *Juncus ensifolius* demonstrated IC_50_ values of 8.25~75.57 toward Hela cells; moreover, after they compared the cytotoxic activities of these compound with non-tumor cells, only Ensifolin A displayed a strong selectivity for Hela cells [42]. These reports are partially consistent with and supporting of our results, namely that compound structures, even with minor differences, would have significant activities towards the same cell lines.

As p53 is essential for HeLa cell apoptosis, and its up expression was measured when treated with compound 5, it is necessary to identify if this up expression is dependent on p53. To do this, three p53 siRNA sequences were screened, and #3 exhibited the most significant inhibition (Appendix A). When single siRNA was performed, the confluence rate increased; when compound 5 was applied, the confluence rate decreased. After normalization, it could be seen that the siRNA + compound 5 group exhibited a better confluence rate than the SiRNA + control group (Figure 8); combined with the survival rate (Figure 9) data, these results indicated that down regulation of p53 could increase the confluence rate and survival rate of HeLa cells treated with compound 5.

The above experimental results are so far new and promising in research on phenanthrene compounds for cervical cancer, and further antitumor reaction mechanisms should be explored in the near future.

It is a long process for a natural product to be developed to a final drug for cancer treatment. All experimental data shown herein exhibited the continuous developing tendency for compound 5. More work should be performed in the near future to take a step further.

## 4. Materials and Methods

### 4.1. General Experimental Procedures

^13^CNMR and ^1^HNMR spectra were recorded on Bruker AV-600 spectrometer. HPLC analyses were carried out using Agilent 1100 series system (Agilent Technologies, Palo, Alto, CA, USA) coupled with a diode array detector. Silica gel (200–300 mesh) was used for column chromatography (CC) and was purchased from Qingdao Marine Chemical Factory (Qingdao, China). HPLC grade methanol, acetonitrile and water were obtained from Kangkede and Wahaha, respectively. The absorbance was measured with a microplate reader (Synergy H1, USA). All analytical grade reagents were purchased from Damao company (Tianjin, China). The apoptotic analysis was carried out using an Annexin V-FITC Apoptosis Detection Kit (Beyotime, Shanghai, China).

### 4.2. Plant Materials

The dried stems of *Dendrobium officinale* were collected from Bozhou herbal medicinal market (Bozhou, China) in 2015. They were identified by Prof. Jin-Cai Lu (School of Traditional Chinese Materia Medica, Shenyang Pharmaceutical University).

### 4.3. Extractions and Isolations

The air-dried stems of *Dendrobium officinale* (5 kg) were extracted with 95% and 85% aqueous ethanol one at a time. The extracted solution was combined and reduced under pressure to yield the crude extract (4.2 L). The same volume of n-BuOH and ethyl acetate was extracted four times, and 110 g and 58 g of n-BuOH and EtOAC extract were obtained, respectively.

The n-BuOH extract was separated using vacuum liquid chromatography (silica gel 200–300 mesh), eluting with CH_2_Cl_2_/MeOH (100:0–1:1, *v*/*v*), and afforded two fractions, A–B. Fraction A was chromatographed on repeated silica gel (PE/EtOAc, 100:0–1:1, *v*/*v*) to generate two fractions (Fr.A1–Fr.A2). Fr. A1 (110 g) was further loaded onto a Sephadex LH-20 column eluted with CH_2_Cl_2_/MeOH (1:1, *v*/*v*) to obtain two fractions (Fr.A1.1–Fr.A1.2). Fr. A1.2 was purified via preparative and semi-preparative HPLC to obtain compounds 4 (2.7 mg), 5 (3.9 mg), 8 (3.7 mg) and 9 (3.7 mg). Similarly, Fr.A2 was further loaded onto a Sephadex LH-20 column eluted with CH_2_Cl_2_/MeOH, (1:1, *v*/*v*) to obtain one fraction (Fr.A2.1); Fr. A2.1 was purified via preparative and semi-preparative HPLC to obtain compound 6 (3.9 mg).

The EtOAC extract was separated via vacuum liquid chromatography (silica gel 200–300 mesh), eluting with CH_2_Cl_2_/MeOH (100:0–1:1, *v*/*v*), and afforded five fractions, C–G. Fraction C (12 g) was chromatographed on repeated silica gel (PE/EtOAc, 100:0–1:1, *v*/*v*) to generate two fractions (Fr.C1–Fr.C2). Fr.C1.1–Fr.C1.4 were obtained via ODS column chromatography, eluted with a gradient of MeO-H_2_O (1:10–1:0) from Fr.C1. Fr.C1.2 was chromatographed on silica gel (cyclohexane-EtOAC) to acquire one fraction (Fr.C1.2.1). Fr.C1.2.1 was purified via preparative and semi-preparative HPLC to obtain compounds 7 (3.4 mg), 2 (7.2 mg), 3 (4.8 mg) and 11 (4.1 mg). Similarly, Fr.C2 was separated using ODS column chromatography, eluted with a gradient of MeO-H_2_O (1:10–1:0), to give one fraction (Fr.C2.1). Fr.C2.1 was purified via HPLC to give compounds 1 (3.1 mg) and 10 (3.7 mg). Fraction F (5 g) was chromatographed on silica gel (CH_2_Cl_2_, Acetone) to generate four fractions (Fr.F1–Fr.F4). Fr.F2 was further submitted to a Sephadex LH-20 column eluted with CH_2_Cl_2_/MeOH (1:1, *v*/*v*) and was purified via preparative and semi-preparative HPLC to obtain compound 12 (13 mg).

### 4.4. Characterization of Compound 5

1,5,6-trimethoxy-2,7-dihydroxyphenanthrene (5): white crystal (MeOH); UV (MeOH) λmax nm (log ε) 212, 232, 286, 295, 309 and 350 (4.35, 4.25, 4.12, 4.00, 3.92 and 3.18); IR (KBr) vmax 3300 cm-1; ^1^H NMR and ^13^C NMR data, see Table 2; EI/MS *m*/*z* (%): 300[M]^+^ (100), 285 (46), 253 (27), 242 (11), 227 (11), 167 (23) and 137 (16) (calculated for C_17_H_16_O_5_, 300.31).

### 4.5. Cell Culture

HeLa and SK-N-As were cultured with MEM and RPMI 1640 complete medium, respectively. MCF-7, Hep G2 and Capan-2 were cultured with DMEM high glucose complete medium. All the media were supplemented with 10% (*v*/*v*) fetal bovine serum (Gibco, New York, NY, USA) and 1% (*v*/*v*) penicillin/streptomycin (100 U/mL penicillin, 10 mg/mL streptomycin) to form a complete medium. Cells were incubated in a humidified atmosphere with 5% CO_2_. Logarithmically growing cells were used in all the experiments.

### 4.6. Cell Viability Assay

The cytotoxic activities of isolated compounds were evaluated using the MTT method with proper modification [40,43]. HeLa cells were seeded in 96-well plates at a density of 5 × 10^3^ cells/well and incubated overnight. Different concentrations (3.125–100 mM) of the tested compounds and positive control were incubated for 48 h. After treatment, 10 μL of MTT (5 mg/mL) was added to each well and incubated for another 3 h. Then, 150 μL/well of DMSO purple formazan crystal solution was supplemented. The cells were measured at 490 nm to detect their absorbance with a microplate reader (Molecular Devices, iMark, San Jose, CA, USA).

### 4.7. Confluence Detection by IncuCyte Living Cell Dynamic Imaging System

The 96-well plates treated with different concentrations (3.125 mM–100 mM) of compound 5 were placed in the IncuCyte Zoom living cell imaging system (Essen BioSciences, Ann Arbor, MI, USA) with proper modifications [44]. The cell images were collected regularly to observe the changes of cell growth dynamically. The time interval of image collection was set to 4 h and the images were collected with a 10× lens. The cell areas in each image were automatically identified by the system’s built-in analysis software. The cell confluence rate under different treatment conditions was calculated with IncuCyte ZOOM 21 CFR Part software Module (Essen BioScience, USA).

### 4.8. Cell Scratch Wound Assay

HeLa cells were plated in 96-well plates at a density of 2 × 10^4^ cells/well and incubated overnight according to previous literature with minor modifications [40]. After the cell confluence rate reached 100%, the wells were collected with a scratcher. When the HeLa cells were treated with indicated compound concentration (0.84 mM), cells were observed and photographed at different time points (0 h, 4 h, 8 h, 12 h, 16 h, 20 h and 24 h) in the IncuCyte living cell imaging system. Cell migration rate at 0 h and 24 h was calculated using IncuCyte Zoom software (Essen BioSciences, USA).

### 4.9. Hoechst 33258 Fluorescence Staining

HeLa cells were cultured in 6-well plates at a density of 15 × 10^4^ cells/well overnight [45]. After 48 h treatment with the compound at different concentrations (3.125 mM, 6.25 mM), 1 mL 4% paraformaldehyde was added and fixed for 15 min at room temperature without light. The fixed solution was blotted out and washed with cold PBS 3 times. A total of 500 μL Hoechst 33258 staining solution was added to each well, after which they were incubated at 37 °C for 15 min in the dark. Then, the fluid was sucked out and the cells were observed under a fluorescent microscope (Nikon, Tokyo, Japan).

### 4.10. Annexin V-FITC/PI Apoptosis Detection

HeLa cells were prepared according to the previous literature with modification [43,46]. After 48 h of treatment with compound 5 at different concentrations (3.125 mM, 6.25 mM, 12.5 mM), cells were harvested and stained with Annexin V-FITC/PI for 15 min at room temperature in the dark and then analyzed using flow cytometry (BIO-RAD, Hercules, CA, USA).

### 4.11. PI Apoptosis Detection and Cell Cycle Detection

HeLa cells were prepared according to the previous literature [43,46]. After 48 h pf treatment with the compound at 3.125 mM concentrations, the supernatant was harvested and fixed with 75% ethanol at 4 °C for at least 2 h. A total of 500 μL propidium iodide staining solution was added to each well. After incubation at 37 °C for 30 min, the cell cycle was detected using flow cytometry (BIO-RAD, USA).

### 4.12. Western Blotting Analysis

HeLa cells were cultured as previous reported [43,47]. After 48 h of treatment with the compound at different concentrations (3.125 mM, 6.25 mM, 12.5 mM), samples were harvested and lysed with RIPA lysis buffer for 30 min. After ultrasonication, cell lysates were centrifuged at 12,000 rpm for 10 min at 4 °C. Afterwards, the protein concentrations were determined using the BCA protein assay kit. The samples were denaturized for 5 min in a 100 °C metal bath. Then, samples were loaded and separated with 12% SDS-PAGE, then transferred onto PVDF membranes for 1.5 h. The PVDF were blocked with 5% skim milk in TBST for 1.5 h at room temperature and incubated with primary antibodies at 4 °C overnight. After washing three times with TBST, the PVDF were incubated with horseradish peroxidase (HRP) conjugated secondary antibody. After washing three times with TBST, the protein bands were detected using ECL luminescence apparatus (BIO-RAD, USA).

### 4.13. Small Interfering RNA (siRNA) Transfection

HeLa cells were prepared as previous reported [48]. Transfection reagents were added when the cell confluence rate reached 50–70%. The jetPrime transfection reagent was used according to the manufacturer’s instructions. For one well, 200 μL JETPrime Buffer and 5 μL siRNA were supplemented. After vortexing for 10 s, 5 μL jetPrime Transfection Reagent was added; after vortexing for another 10 s, the plates were incubated at room temperature for 10 min. After 16 h, the transfected cells were collected, recounted and inoculated into 96-well plates. After 10 h, the cells were treated with the compound (3.125 μM). After 48 h, MTT detection was performed. The 96-well plate was placed in the incubator to analyze cell survival by taking timed images from the IncuCyte Zoom living cell imaging system.

### 4.14. Statistics

All results and data were confirmed in at least three separate experiments. Statistical comparisons were analyzed via one-way ANOVA using GraphPad Prism from GraphPad Software (GraphPad software 8.0.2, San Diego, CA, USA). The experimental data are expressed as mean ± standard deviation (X ± SD). * *p* < 0.05 was considered statistically significant.

## Figures and Tables

**Figure 1 ijms-24-15375-f001:**
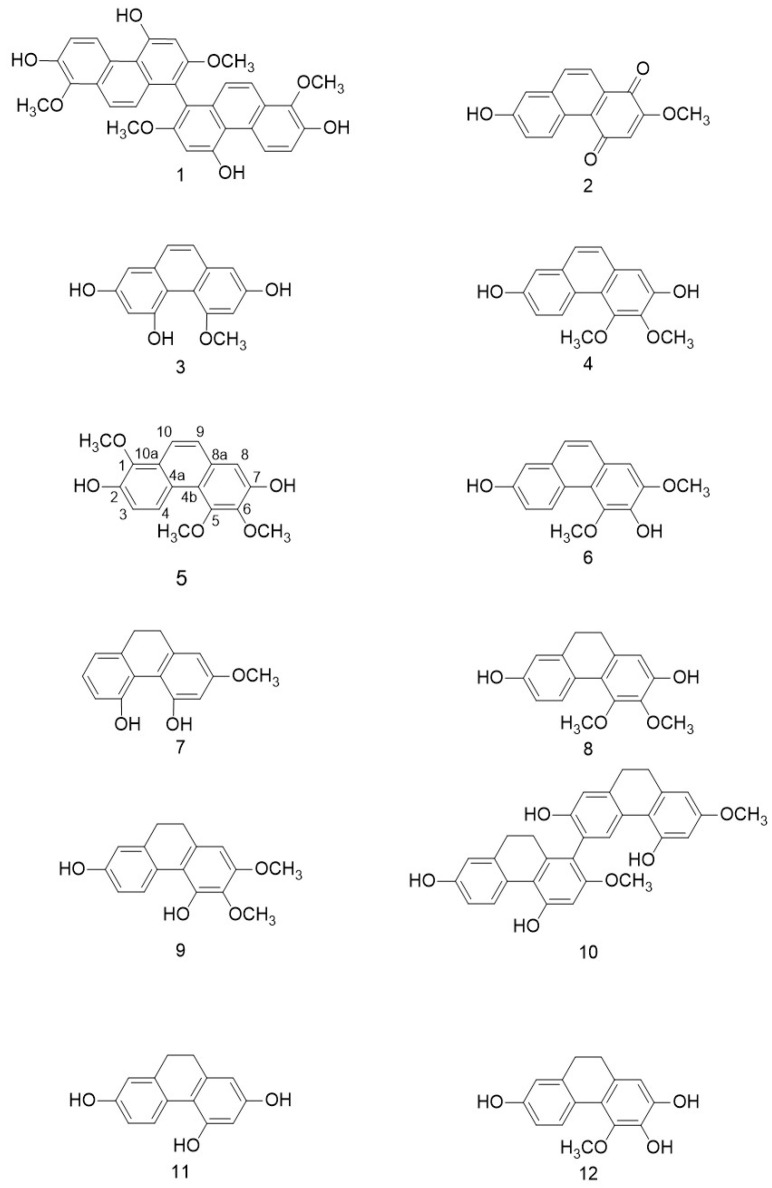
The chemical structures of compounds 1–12.

**Figure 2 ijms-24-15375-f002:**
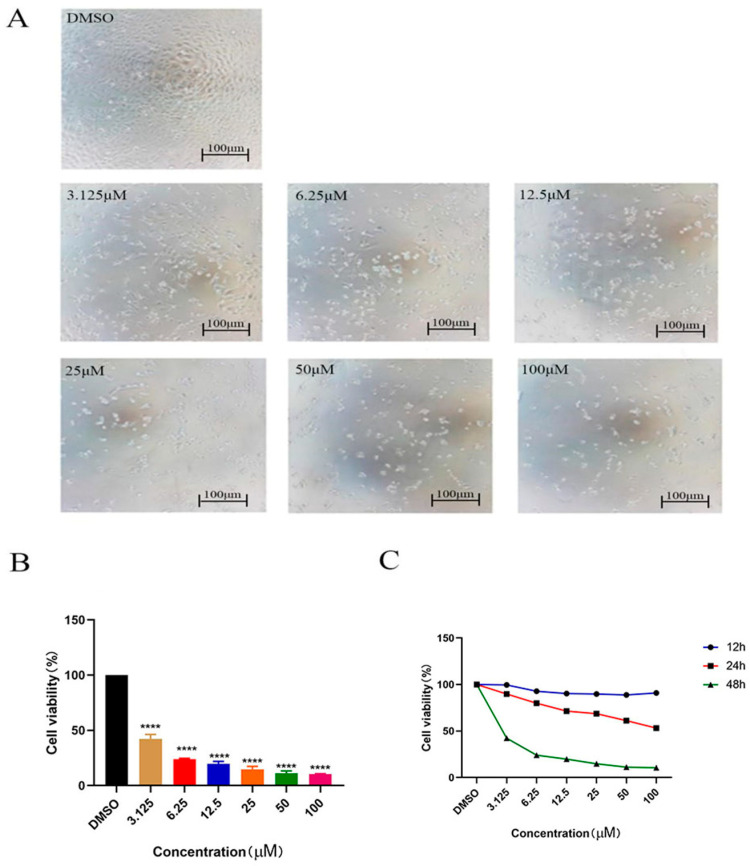
Cell morphological and survival rate of compound 5 towards HeLa Cells. (**A**) Microscopic morphological data of HeLa cells treated with different concentrations of compound 5 for 48 h. (**B**) Survival rate of HeLa cells treated with different concentrations of compound 5 for 48 h. (**C**) Survival rate of HeLa cells treated with different concentrations of compound 5 at different times. The statistical results are expressed as the mean ± SD, *n* = 3, **** *p* < 0.0001 vs. control group.

**Figure 3 ijms-24-15375-f003:**
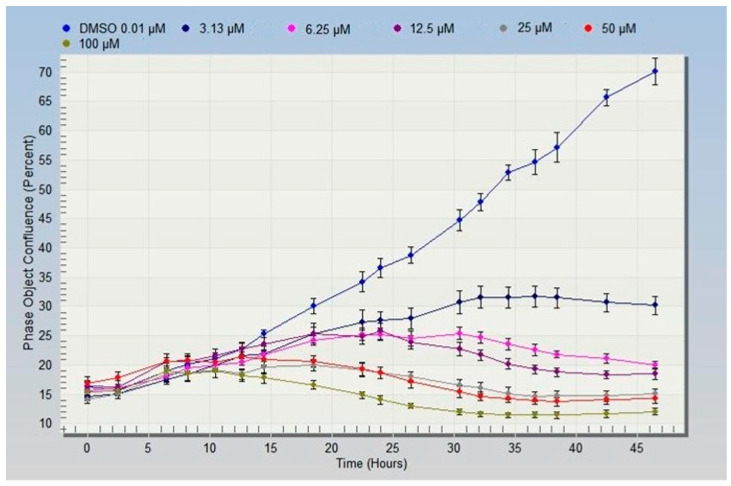
The confluence curve for HeLa cells treated with different concentrations of compound 5.

**Figure 4 ijms-24-15375-f004:**
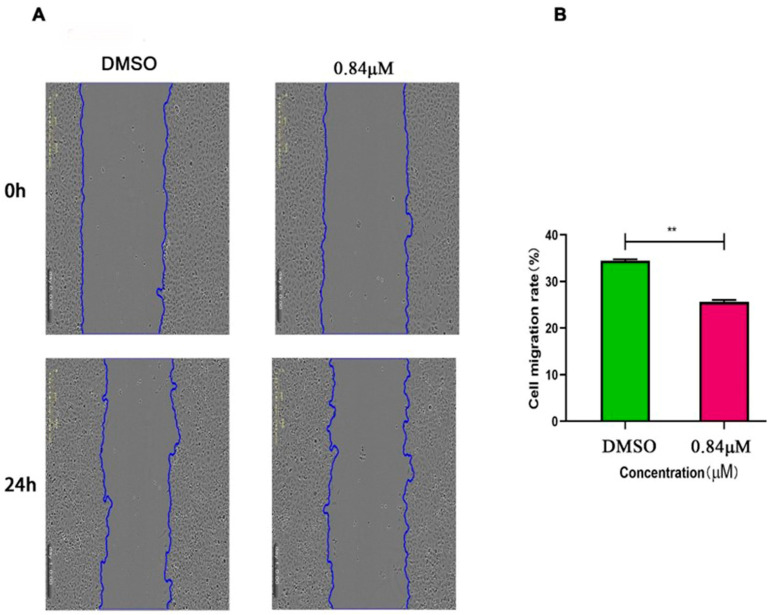
Compound 5 inhibited the migration of HeLa cells. (**A**) The migration of HeLa cells under the microscope after treatment with DMSO and compound 5 for 0 h and 24 h. (**B**) The average migration rate of HeLa cells after treatment with compound 5 for 24 h. The statistical results are expressed as the mean ± SD, *n* = 3, ** *p* < 0.01 vs. control group.

**Figure 5 ijms-24-15375-f005:**
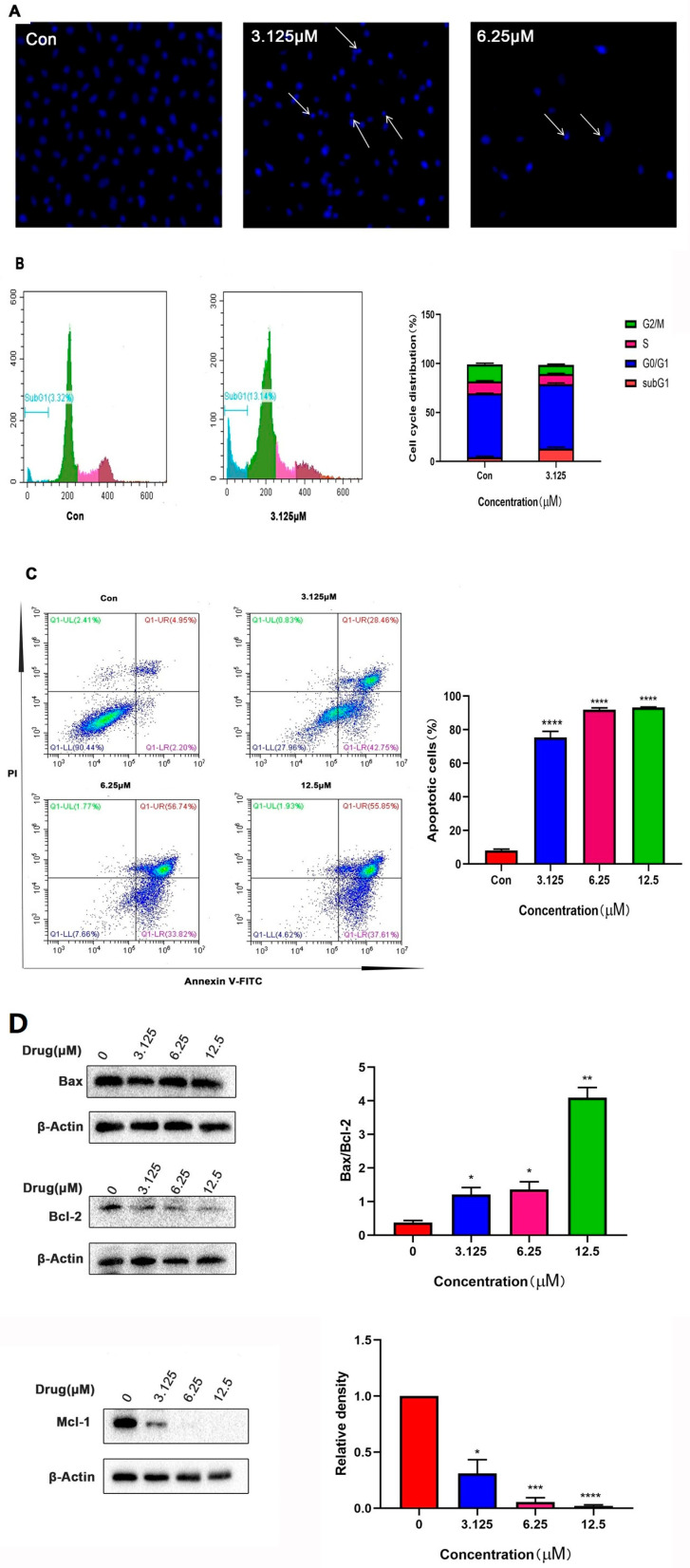
Compound 5 could induce apoptosis in HeLa cells. (**A**) Hoechst 33258 stain on HeLa nucleus under fluorescence microscope. (**B**) PI staining and flow cytometry to evaluate the apoptotic ratio of compound 5-treated HeLa cells. (**C**) Annexin V-PI staining was used to determine the apoptotic ratio of compound 5-treated HeLa cells. (**D**) Apoptosis-associated proteins were detected via Western blot assay. The relative levels of Bax, Bcl2 and MCL-1 were quantified with Graphpad 8.0.2. * *p*< 0.05, ** *p* < 0.01, *** *p* < 0.001, **** *p* < 0.0001 vs. control group.

**Figure 6 ijms-24-15375-f006:**
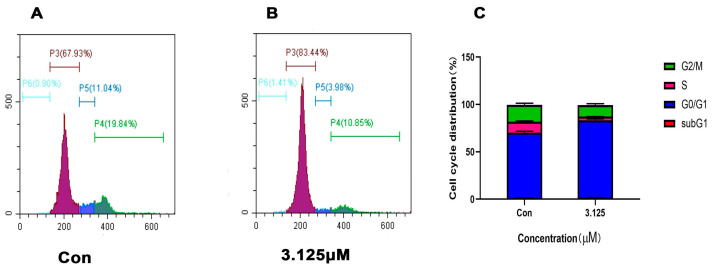
Compound 5 could block HeLa cell cycle at G0/G1. (**A**,**B**) Cell cycle was detected by PI single staining and flow cytometry. (**C**) The proportion of each cell cycle after treatment with compound 5 for 36 h. The statistic results were expressed as the mean ± SD, n = 3.

**Figure 7 ijms-24-15375-f007:**
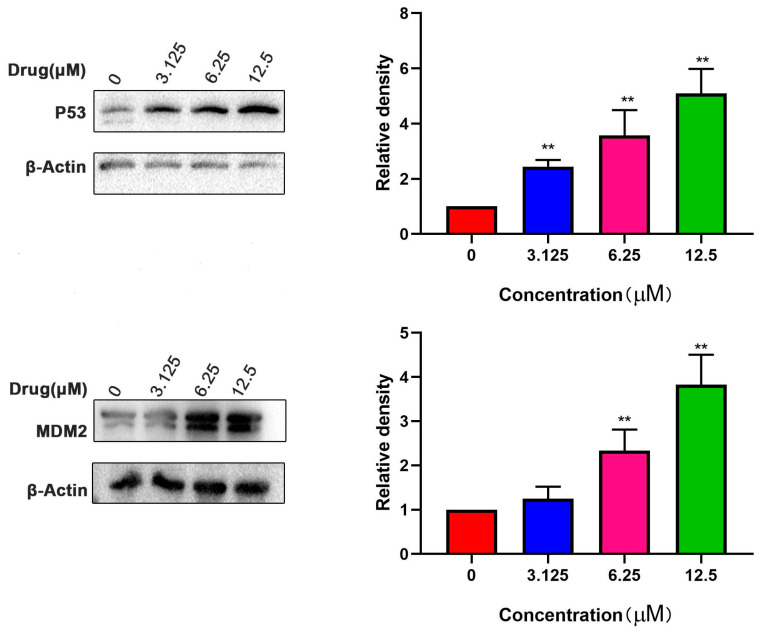
Compound 5 promoted the expression of P53 and MDM2 protein in HeLa cells. ** *p* < 0.01 vs. control group.

**Figure 8 ijms-24-15375-f008:**
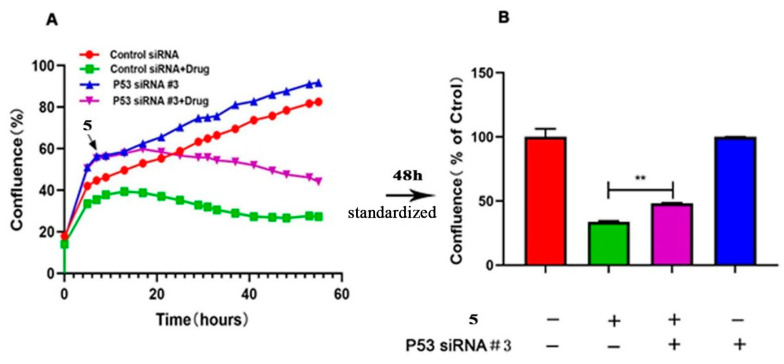
P53 mitigates compound 5’s inhibition of the proliferation of HeLa cells. (**A**) The confluence rate curve of HeLa cells after transfection with P53 siRNA and treatment with compound 5 for 48 h. (**B**) The confluence rate of HeLa cells after transfection with P53 siRNA and treatment with compound 5 for 48 h. The statistical results are expressed as the mean ± SD, *n* = 3, ** *p* < 0.01 vs. control group.

**Figure 9 ijms-24-15375-f009:**
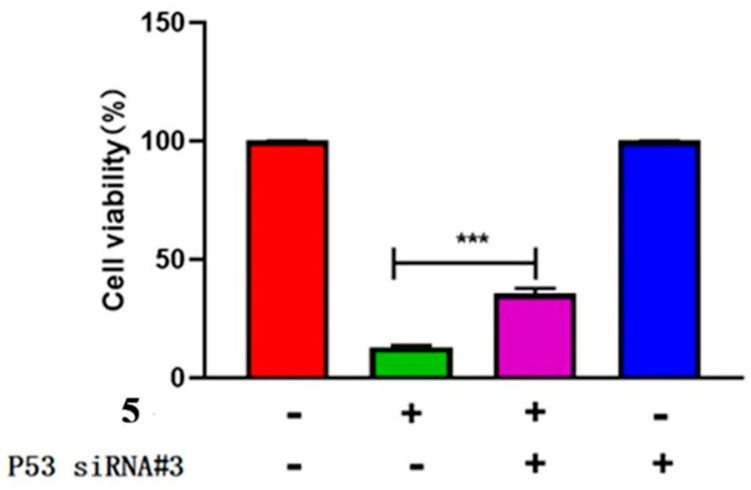
The survival rate of HeLa cells after transfection of P53 siRNA and treatment with compound 5 for 48 h. The statistical results are expressed as the mean ± SD, *n* = 3, *** *p* < 0.001 vs. control group.

**Table 1 ijms-24-15375-t001:** Effects of 12 phenanthrene compounds on cell survival after 48 h of HeLa, MCF7, SK-N-As, Capan-2 and Hep G2.

Compounds			IC_50_ Values (μM)		
	HeLa (Cervical Cancer)	MCF7 (Breast Cancer)	SK-N-AS (Neuroblastoma)	Capan-2 (Pancreatic Cancer)	Hep G2 (Hepatoma Cancer)
1	21.36 ± 3.05	26.73 ± 1.88	25.87 ± 4.03	26.29 ± 0.24	30.85 ± 2.46
2	54.56 ± 0.97	25.67 ± 3.04	25.60 ± 3.45	14.96 ± 0.61	10.87 ± 0.55
3	>200	>200	>200	>200	>200
4	113.47 ± 5.32	180.05 ± 3.75	>200	>200	>200
5	0.42 ± 0.57	61.29 ± 6.01	49.89 ± 5.72	68.21 ± 0.76	0.20 ± 0.12
6	42.20 ± 0.44	>200	>200	>200	183.3 ± 2.05
7	>200	>200	>200	>200	>200
8	>200	>200	>200	>200	174.2 ± 12.35
9	90.66 ± 5.40	>200	>200	>200	>200
10	58.68 ± 1.68	100.1 ± 0.57	>200	>200	>200
11	>200	>200	>200	>200	>200
12	73.91 ± 1.74	91.43 ± 6.20	100.66 ± 3.35	72.83 ± 1.85	>200
Cisplatin	7.84 ± 1.01	2.75 ± 1.58	13.99 ± 3.27	24.97 ± 4.13	2.28 ± 0.36
Paclitaxel	2.01 ± 1.0	6.77 ± 1.76	7.24 ± 0.20	76.64 ± 9.75	0.21 ± 0.07

**Table 2 ijms-24-15375-t002:** ^1^H and ^13^C NMR data of compound 5.

NO.	1H-NMR (J in Hz)	13C-NMR
1	-	143.4
2	-	147.5
3	7.17 (1H, d, J = 9.3 Hz, H-3)	118.2
4	9.09 (1H, d, J = 9.3 Hz, H-4)	124.3
4a		130.7
5	-	152.8
4b		119.5
6	-	142.7
7	-	150.4
8	7.07 (1H, s, H-8)	110.1
8a		128.4
9	7.52 (1H, d, J = 9.0 Hz, H-9)	127.8
10	7.87 (1H, d, J = 9.0 Hz, H-10)	120.5
10a		125.5
1-OCH_3_	4.00 (3H, s)	61.4
5-OCH_3_	3.93 (3H, s)	61.4
6-OCH_3_	3.91 (3H, s)	60.4

^1^H NMR at 600 MHz, ^13^C NMR at 150 MHz, obtained in CD_3_OD.

## Data Availability

The data presented in this study are available on reasonable request from the corresponding author.

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
