# Peer review of "1,5,6-Trimethoxy-2,7-dihydroxyphenanthrene from Dendrobium officinale Exhibited Antitumor Activities for HeLa Cells"

_ijms, 2023, doi:10.3390/ijms242015375_

Round 1

Reviewer 1 Report

The authors brought results contributing to further knowledge of the active substances of dendrobium officinale. Unfortunately, the article suffers from several serious flaws that prevent its publication.

The major flaws are:

The structure of the studied substances is not sufficiently proven. Spectral characterization is missing for all substances except for substance 5. However, in the case of substance 5, the methoxy groups are omitted in the spectrum recording (Table 2) and the four internal carbons of phenanthrene are missing in the carbon spectrum. Moreover, a more detailed 2D NMR analysis demonstrating the exact position of the methoxy groups and hydroxyls is missing.

The cited literature does not contain the data to which the article refers. The literature listed in the References section is apparently quite different from that referred to in the text - for example, right at the beginning in references 1-9 we do not find the data given in the section. What is more serious, the literature cited in section 2.1 also does not refer to the structural characteristics of the studied substances.

The achieved results are not critically evaluated. Part 3 Discussion and conclusion does not include a comparison of the cytotoxic activity of structurally similar substances from other species, nor does it attempt to clarify the completely different activities of very similar substances within the studied group.

The small amount (3.9 mg) of isolated substance 5 raises another doubt. It is hard to believe that such an amount was sufficient to carry out all the described experiments, some of which were carried out in relatively high concentrations.

Some experiments are not described in sufficient detail: The origin of cell lines is missing. The solvents used to dissolve the tested substances are missing. Standardized procedures for methods 4.6 – 4.13 should be described in detail, or a reference to a detailed description in the literature, for example, the instrument manufacturer's manual, should be given. Not all devices are unambiguously identified (type, manufacturer, equipment).

Some paragraphs are confused or redundant:

lines 45 - 53 should not be in Introduction section, because they summarize the results.

line 104 excessive

lines 220 – 225 are in the wrong place

Author Response

     Thank you very much for your email and reviewer's comments on our manuscript entitled " 1,5,6-trimethoxy-2,7-dihydroxyphenanthrene from Dendrobium officinale exhibited antitumor activities for HeLa cells". Your comments are very helpful to revise and improve our manuscript. After carefully studied our reviewers' comments, we have made corresponding corrections with the track mark in the manuscript. Also, we’ve listed our answers to the reviewer's comment as below in blue italic font. Hope these efforts will make the manuscript acceptable for publication.

     The main corrections in the paper and the responses to editor’s comments are as follows:

The authors brought results contributing to further knowledge of the active substances of dendrobium officinale. Unfortunately, the article suffers from several serious flaws that prevent its publication.

The major flaws are:

The structure of the studied substances is not sufficiently proven. Spectral characterization is missing for all substances except for substance 5. However, in the case of substance 5, the methoxy groups are omitted in the spectrum recording (Table 2) and the four internal carbons of phenanthrene are missing in the carbon spectrum. Moreover, a more detailed 2D NMR analysis demonstrating the exact position of the methoxy groups and hydroxyls is missing.

Thanks for your kind reviewing, we’ve renumbered the compound 5 and modified the data in Table 2, to provide the sufficient support of structure compound 5. As the other 11 compounds are known compounds, we just compare the spectra data with the published references and identified them. So, we did not list those spectra data in the manuscript.

The cited literature does not contain the data to which the article refers. The literature listed in the References section is apparently quite different from that referred to in the text - for example, right at the beginning in references 1-9 we do not find the data given in the section. What is more serious, the literature cited in section 2.1 also does not refer to the structural characteristics of the studied substances.

Please pardon our careless mistake for the reference list. After careful inspection of the whole texture, all references been updated in the revised manuscript.

The achieved results are not critically evaluated. Part 3 Discussion and conclusion does not include a comparison of the cytotoxic activity of structurally similar substances from other species, nor does it attempt to clarify the completely different activities of very similar substances within the studied group.

Thanks for pointing out this. We’ve supplemented some comparison of the cytotoxic activity of structurally similar substances from other species, such as the Phenanthrenes from Juncus species (Ref 40-42). These literatures reported that several compounds also exhibited small IC50 towards Hela Cell, which supported our results, that the some phenanthrenes are quite positive for the Hela cell. As our reviewers indicated, the reason why these completely different activities among very similar substances were significantly depended with their structures. It is not easy to provide the extensive and detailed explanations, we are going to perform more experiments and molecular dynamics to explore their relationship. So we could not provide them now. Sorry for this.

The small amount (3.9 mg) of isolated substance 5 raises another doubt. It is hard to believe that such an amount was sufficient to carry out all the described experiments, some of which were carried out in relatively high concentrations.

This amount is the data for one batch preparation. In our experimental process, many batches of extraction and preparation were performed, so that the compounds amounts are enough for the cytotoxic experiment research.

Some experiments are not described in sufficient detail: The origin of cell lines is missing. The solvents used to dissolve the tested substances are missing. Standardized procedures for methods 4.6 – 4.13 should be described in detail, or a reference to a detailed description in the literature, for example, the instrument manufacturer's manual, should be given. Not all devices are unambiguously identified (type, manufacturer, equipment).

The Hela, Hep G2 and Capan-2 cell lines were purchased from the National Collection of Authenticated Cell Cultures (Shanghai, China). The MCF-7 and SK-N-AS cell lines were kindly obtained from Prof. Zhijie Li of China Medical University.

All the solvents used to dissolve the tested compounds are DMSO.

Standardized procedures for methods 4.6-4.13 were rephrased, supplemented with the device descriptions, and enclosed the reference literatures (Ref 40-48). 

Some paragraphs are confused or redundant:

lines 45 - 53 should not be in Introduction section, because they summarize the results.

Lines 45-53 had been deleted from the introduction section.

line 104 excessive

Line 104 had been deleted.

lines 220 – 225 are in the wrong place

Line 220-225 had been deleted.

Reviewer 2 Report

The manuscript entitled “1,5,6-trimethoxy-2,7-dihydroxyphenanthrene from Dendrobium officinale exhibited antitumor activities for HeLa cells (Manuscript ID:  ijms-2587315) by Zhang et al. describes the isolation and characterisation of 12 phenanthrene compounds and their evaluation against five different cancer cell lines. They have also performed the mechanistic studies for the most potent compound. However, the manuscript needs major changes before accepted for publication.

1.      The article needs rearrangements of content in different sections.

a.       The introduction section details about the results and conclusions of the study, and discussion and conclusion contain natural products introduction, which can be rearranged.

b.      The introduction section requires corrections, the following sentence needs completion “According to the report of the International Agency for Research on Cancer” (Line 28)

c.       Dendrobium officinale should be italicized in the entire manuscript.

d.      All the subscripts (IC50), symbols, (µM), spacings (h, °C), should be corrected in the entire manuscript.

e.       Spelling mistakes need to be corrected in the entire manuscript. Line 286, scratch, and many like these!

f.        The use of term “ingredients” is not going well with the scientific nature of the manuscript. Therefore, I suggest the authors to replace with compounds or phenanthrenes or anything that best suits the manuscript.

g.      Line 85, this can be consistent with apoptosis results but not coincident, to my understanding.

h.      The images were hyperlinked to some D drive in other language, make sure those hyperlinks are removed from the Figures.

i.        Language polishing is required.

2.      From the manuscript, it is understood that IC50 was used as the criteria in selecting the lead compound for performing mechanistic studies. But what is the reason for choosing HeLa cell line over Hep G2 cells, when compound 5 exhibited better IC50 in Hep G2?

3.      When authors performed extensive spectroscopic analyses (Line 58), why is the spectral data or the spectra not provided in the manuscript, other than for compound 5?

4.      All the structures in Figure 1 needs to be cleaned and the compound 5 numbering needs attention. Why authors numbered OH as 16, and one OCH3 as 17?

5.      Any methods used in the study needs citation, So I would suggest the authors to cite the following article for MTT assay both in the results (Line 72) and experimental section, (Line 271), and include other references appropriately for cell migration and cell cycle etc based on literature.

Tokala R, Thatikonda S, Vanteddu US, Sana S, Godugu C, Shankaraiah N. Design and Synthesis of DNA-Interactive β-Carboline-Oxindole Hybrids as Cytotoxic and Apoptosis-Inducing Agents. ChemMedChem. 2018, 13(18):1909-1922. doi: 10.1002/cmdc.201800402.”

6.      Table 1 should have a legend describing the cell lines (type of cancer) and other details.

7.      Table 1. Did authors tried testing the phenanthrenes against any normal human cell line to understand the selectivity towards cancer cells?

8.      Figure 4A has an extra 5 (µM) on top and its misleading, make appropriate changes.

9.      Figure 4, any specific reason in choosing to double the IC50 concentration, but the migration differences are not so great with respect to DMSO treated cells.

10.  What is the difference between section 2.5 and individual cell cycle studies? Did authors want to make any specific conclusion from both cell cycles or whether it is all same?

11.   Did authors conclude anything on dose dependency in these mechanistic studies?

12.  Table 2. Where is the spectral data about methoxy groups?

13.  In section 4.6, Line 272, is it 5 × 103 cells/well? Section 4.8, Line 287, is it 2 × 104 cells/well? Cross-check the procedures for all the sections and make the necessary corrections in the entire manuscript.

14.  References need attention, for example, Ref 21, 22, 23, 29, 32, 35: page numbers missing. All the references need to be uniformly formatted.

   Language polishing is suggested.

Author Response

     Thank you very much for your email and reviewer's comments on our manuscript entitled " 1,5,6-trimethoxy-2,7-dihydroxyphenanthrene from Dendrobium officinale exhibited antitumor activities for HeLa cells". Your comments are very helpful to revise and improve our manuscript. After carefully studied our reviewers' comments, we have made corresponding corrections with the track mark in the manuscript. Also, we’ve listed our answers to the reviewer's comment as below in blue italic font. Hope these efforts will make the manuscript acceptable for publication.

     The main corrections in the paper and the responses to editor’s comments are as follows:

The manuscript entitled “1,5,6-trimethoxy-2,7-dihydroxyphenanthrene from Dendrobium officinale exhibited antitumor activities for HeLa cells” (Manuscript ID: ijms-2587315) by Zhang et al. describes the isolation and characterisation of 12 phenanthrene compounds and their evaluation against five different cancer cell lines. They have also performed the mechanistic studies for the most potent compound. However, the manuscript needs major changes before accepted for publication.

  1. The article needs rearrangements of content in different sections.
  2. The introduction section details about the results and conclusions of the study, and discussion and conclusion contain natural products introduction, which can be rearranged.

Thanks for your kind reviewing and suggestion. Both the introduction and discussion section had been rearranged.

  1. The introduction section requires corrections, the following sentence needs completion “According to the report of the International Agency for Research on Cancer” (Line 28)

Yes, the period symbol had been changed to comma symbol. The sentence had been completed.

  1. Dendrobium officinale should be italicized in the entire manuscript.

Yes, all the Dendrobium officinale in the main text had been italicized.

  1. All the subscripts (IC50), symbols, (μM), spacings (h, °C), should be corrected in the entire manuscript.

Yes, all the subscripts, symbols, μM and spacing had been corrected.

  1. Spelling mistakes need to be corrected in the entire manuscript. Line 286, scratch, and many like these!

Yes, the spelling mistakes had been corrected for the entire manuscript.

  1. The use of term “ingredients” is not going well with the scientific nature of the manuscript. Therefore, I suggest the authors to replace with compounds or phenanthrenes or anything that best suits the manuscript.

Yes, the word “ingredient” had been replaced with the term “compound/phenanthrene”.

  1. Line 85, this can be consistent with apoptosis results but not coincident, to my understanding.

We agree with your opinion, and we rephrased the sentences as “This is consistent with the apoptosis results.”

  1. The images were hyperlinked to some D drive in other language, make sure those hyperlinks are removed from the Figures.

All the hyperlinks had been deleted.

  1. Language polishing is required.

Yes, we’ve found some native speaker for polishing this revised manuscript.

  1. From the manuscript, it is understood that IC50 was used as the criteria in selecting the lead compound for performing mechanistic studies. But what is the reason for choosing HeLa cell line over Hep G2 cells, when compound 5 exhibited better IC50 in Hep G2?

Thanks for pointing this out. In our experimental procedures, the Hela cells were firstly tested, and the promising results encouraged us to continue the further studies, and extended the candidate cell lines to Hep G2 and others. For the time limitation, in this manuscript we only summarized and reported the Hela cell data and results. We are still working on the Hep G2 cells and would like to summarize and submit those data in the near future.

  1. When authors performed extensive spectroscopic analyses (Line 58), why is the spectral data or the spectra not provided in the manuscript, other than for compound 5?

Thanks for your kind reviewing. The 12 compounds obtained in this manuscript are not new compounds, so we compared their spectra data with the references to identified their structures.

  1. All the structures in Figure 1 needs to be cleaned and the compound 5 numbering needs attention. Why authors numbered OH as 16, and one OCH as 17?

Thanks for your kind reviewing. The numbering for compound 5 had been modified in Figure 1.

  1. Any methods used in the study needs citation, So I would suggest the authors to cite the following article for MTT assay both in the results (Line 72) and experimental section,(Line 271), and include other references appropriately for cell migration and cell cycle etc based on literature.

“Tokala R, Thatikonda S, Vanteddu US, Sana S, Godugu C, Shankaraiah N. Design andSynthesis of DNA-Interactive β-Carboline-Oxindole Hybrids as Cytotoxic and Apoptosis-Inducing Agents. ChemMedChem. 2018, 13(18):1909-1922. doi:10.1002/cmdc.201800402 (https://doi.org/10.1002/cmdc.201800402).”

Thanks for your kind help. The above literature and other references had been supplemented for the experiment section(Ref 40-48).

  1. Table 1 should have a legend describing the cell lines (type of cancer) and other details.

Thanks for your kind suggestion. We’ve supplemented a brief description in the parentheses to indicate the type of cancer.

  1. Table 1. Did authors tried testing the phenanthrenes against any normal human cell line to understand the selectivity towards cancer cells?

There are other literatures indicated the data of phenanthrenes against some normal human cell lines, Ref[40-42], so we did not perform this experiment.

  1. Figure 4A has an extra 5 (μM) on top and its misleading, make appropriate changes.

In the updated figure 4A, the 5(μM) had been deleted.

  1. Figure 4, any specific reason in choosing to double the IC concentration, but the migration differences are not so great with respect to DMSO treated cells.

We’ve tried several concentrations to perform this test, and the 2X IC50 concentration demonstrated the difference between the experimental and control group, that is why the double IC50 concentration was used. And it is true that the visual difference is not so significant, despite the data supported so.

Also, this test is performed and finished within 24 hours to guarantee the cells maintain alive, if the culture time could be extended, the migration range may be more significant.

  1. What is the difference between section 2.5 and individual cell cycle studies? Did authors want to make any specific conclusion from both cell cycles or whether it is all same?

Actually, all the experiments performed in section 2.5 are aiming to one conclusion, i.e. the tested compounds could induce apoptosis in Hela cells. Both the PI staining (cell cycle) and Annexin V-FITC/PI staining (cell apoptosis) simultaneously proved the compound could induce the apoptosis of Hela cells, these two experimental results reinforced the same conclusion.

  1. Did authors conclude anything on dose dependency in these mechanistic studies?

Yes, in the apoptosis and western blotting experiments, the dose dependency could be concluded.

  1. Table 2. Where is the spectral data about methoxy groups?

We’ve supplemented the spectral data of methoxy group in the table 2.

  1. In section 4.6, Line 272, is it 5 × 103 cells/well? Section 4.8, Line 287, is it 2 × 104cells/well? Cross-check the procedures for all the sections and make the necessary corrections in the entire manuscript.

We’ve checked the cell amounts, and supplemented the reference to support our descriptions.

  1. References need attention, for example, Ref 21, 22, 23, 29, 32, 35: page numbers missing. All the references need to be uniformly formatted.

Thanks for your kind reviewing. We’ve checked and updated all the references.

Language polishing is suggested.

We’ve found some native speaker to help us polishing this manuscript.

Reviewer 3 Report

In this study, Liang et al. characterised 12 phenanthrene compounds from Dendrobium officinale and studied their In vitro anti-cancer activity using various tumor cell lines. The work is interesting.

Comments:

1.                   “the most significant cytotoxic effect against HeLa and Hep G2 cells, with an IC50 of 0.42 and 0.20 µM”. Do you know if this compound has less toxic effect towards corresponding normal cells (or immortalised cell lines)?  

2.                   Under 2.2. MTT assay, have you tried to use 3D culture to see to anti-cancer activity in 3D spheroids?

3.                   “In this study, 12 phenanthrene compounds were extracted and isolated from Dendrobium officinale”. This might not be right. You might have extracted more than 12 phenanthrene compounds, and some may below detection limit or not identified by limited techniques used.

4.                   Figure 4. control group treated with DMSO; the level of DMSO should be given.

5.                   Figure 7, top panel, it seems that the densities of actin bands are not even.

6.                   Chang uM to µM.

Please check the grammars and word used in the Abstract.

Author Response

     Thank you very much for your email and reviewer's comments on our manuscript entitled " 1,5,6-trimethoxy-2,7-dihydroxyphenanthrene from Dendrobium officinale exhibited antitumor activities for HeLa cells". Your comments are very helpful to revise and improve our manuscript. After carefully studied our reviewers' comments, we have made corresponding corrections with the track mark in the manuscript. Also, we’ve listed our answers to the reviewer's comment as below in blue italic font. Hope these efforts will make the manuscript acceptable for publication.

     The main corrections in the paper and the responses to editor’s comments are as follows:

In this study, Liang et al. characterised 12 phenanthrene compounds from Dendrobium officinale and studied their In vitro anti-cancer activity using various tumor cell lines. The work is interesting.

Comments:

  1. “the most significant cytotoxic effect against HeLa and Hep G2 cells, with an IC50 of 0.42 and 0.20 μM”. Do you know if this compound has less toxic effect towards corresponding normal cells (or immortalised cell lines)?

Sorry, in this experiment, we did not perform the tests for the normal cells. In our future work, this section would be considered and performed.

  1. Under 2.2. MTT assay, have you tried to use 3D culture to see to anti-cancer activity in 3D spheroids?

Sorry, we did not consider the 3D spheroid for the MTT assay.

  1. “In this study, 12 phenanthrene compounds were extracted and isolated from Dendrobium officinale”. This might not be right. You might have extracted more than12 phenanthrene compounds, and some may below detection limit or not identified by limited techniques used.

We agree with your opinion. In this study, these 12 compounds are those could be extracted and isolated, so we just reported them as what we’ve done. Surely there are other compounds whose concentration is lower than the detection limit so that could not be collected and enriched for further identification.

  1. Figure 4. control group treated with DMSO; the level of DMSO should be given.

The DMSO concentration is 0.2%.

  1. Figure 7, top panel, it seems that the densities of actin bands are not even.

In the original graphics, the densities of actin are almost even. Please check the original data for your reference.

  1. Chang uM to μM.

Yes, all the uM had been changed to μM.

Please check the grammars and word used in the Abstract

Yes, the grammars and words in Abstract section had been checked and rephrased.

Thanks again for your reviewing.

Round 2

Reviewer 2 Report

The revised manuscript entitled “1,5,6-trimethoxy-2,7-dihydroxyphenanthrene from Dendrobium officinale exhibited antitumor activities for HeLa cells (Manuscript ID:  ijms-2587315) by Zhang et al. describes the isolation and characterization of 12 phenanthrene compounds and their evaluation against five different cancer cell lines. The authors incorporated the suggestions and improved the quality of the manuscript. However, the manuscript needs these minor corrections before being accepted for publication.

1.       “According to the report of the International Agency for Research on Cancer” (Line 28), the statistics of cancer cases for 2020 are already available, include them rather than the estimated values.

2.      “The numbering for compound 5 had been modified in Figure 1.” Yet, numbering of the compound 5 needs attention in Figure 1. Why are substitutions like methoxy, and hydroxyl groups numbered?

3.      References need attention, for example,

a.       References 15, 32, 40, 41: page numbers missing.

b.      Reference 20: Title is in italics.

c.       Reference 18.

d.      All the references need to be uniformly formatted.

None

Author Response

     Thank you again for your reviewing. We’ve modified the manuscript according to your comments as below.

  1. “According to the report of the International Agency for Research on Cancer” (Line 28), the statistics of cancer cases for 2020 are already available, include them rather than the estimated values.

Thanks for pointing this. We’ve modified the sentence in Line 28.

  1. “The numbering for compound 5 had been modified in Figure 1.” Yet, numbering of the compound 5 needs attention in Figure 1. Why are substitutions like methoxy, and hydroxyl groups numbered?

Thanks for pointing this. We’ve deleted the number of the substitutions and updated the Figure 1.

  1. References need attention, for example,
    1. References 15, 32, 40, 41: page numbers missing.
    2. Reference 20: Title is in italics.
    3. Reference 18.
    4. All the references need to be uniformly formatted.

All the references pages and formats had been checked and updated.

Thank you for your quality time.

Reviewer 3 Report

I recommend to accept this revised version for publication in IJMS.

Author Response

Thank you again for your reviewing and approved our manuscript.

Your work were highly appreciated.